# Occurrence climatology of equatorial plasma bubbles derived using FormoSat-3/COSMIC GPS radio occultation data

Ankur Kepkar[1,2], Christina Arras[2], Jens Wickert[1,2], Harald Schuh[1,2], Mahdi Alizadeh[1,3], and Lung-Chih Tsai[4]

[1]Technische Universität Berlin, Germany
[2]German Research Centre for Geosciences GFZ, Potsdam, Germany
[3]K.N. Toosi University of Technology, Tehran, Iran
[4]National Central University, Chung-Li, Taiwan

**Correspondence:** Ankur Kepkar (kepkar@gfz-potsdam.de)

**Abstract.** The Global Positioning System - Radio Occultation (GPS-RO) technique is used to comprehend the global distribution of equatorial plasma bubbles, which are characterized by depletion regions of plasma in the F-region of the ionosphere. Their occurrence climatology is derived using GPS-RO data from FormoSat-3/COSMIC between 2007 and 2017. Plasma bubbles that cause intense scintillation on the radio signals are identified based on the S4 index derived from the 1 Hz raw signal-to-noise ratio measurements. The analyses revealed that bubbles biased by background plasma density, which occur along the geomagnetic equator, had an occurrence peak around the dip equator during high solar activity. Moreover, the peak shifted between Africa and America depending on different solar conditions. Furthermore, plasma bubbles developed around 19:00 Local Time (LT) with maximum occurrence around 21:00 LT during solar maximum and $\sim 22:00$ LT during solar minimum. The occurrence of bubbles showed a strong dependence on longitude, season, and solar cycle with the peak occurrence rate in the African sector during March equinox during high solar activity, which appeared in congruence with the previous studies. The GPS-RO technique allowed an extended analysis on the altitudinal distribution of global equatorial plasma bubbles obtained from high vertical resolution profiles. Thus, making it a convenient tool, which could be further used with other techniques to provide a comprehensive view of such ionospheric irregularities.

## 1 Introduction

The Equatorial Plasma Bubbles (EPBs) are large regions of plasma depletion, which are prominent in the F-region of the ionosphere. These EPBs generally exist in clusters (Singh et al., 1997) and often deter the radio waves (e.g., GPS signals) penetrating through it, causing serious implications on its applications. These plasma bubbles primarily occur at low latitudes and induce rapid fluctuation in the amplitude as well as phase of the radio signals. This distortion of the signals is often termed as scintillation (Yeh and Liu, 1982). EPBs are also known by its generic name as Equatorial Spread F (ESF), which

are perceived as a spread or diffused echoes in the ionosonde readings (Booker and Wells, 1938; Whalen, 1997). Besides, they appear as plume-like structures in radar observations (Kudeki and Bhattacharyya, 1999) and emission depletions in airglow images (Sahai et al., 2000).

EPBs are a night-time phenomenon and are initiated through the Rayleigh-Taylor Instability (RTI) mechanism in the bottomside of the F-region (Sultan, 1996; Woodman, 2009). Various theories related to seed perturbation like atmospheric gravity waves (AGWs) as well as vertical shear of zonal plasma drift are considered amongst the likely source to trigger the RTI mechanism (Kudeki et al., 2007; Abdu et al., 2009; Huang et al., 2011; Taori et al., 2011). Other than these seed sources, off-equatorial ionospheric phenomena such as sporadic-E layers and medium-scale traveling ionospheric disturbances have also been contemplated for possible seed activity along the equipotential magnetic field lines (Abdu et al., 2003; Tsunoda, 2007). However, AGWs with wavelengths larger than 100 km seed equatorial plasma bubble by causing perturbation in the lower thermosphere, i.e., E-region, which then maps it onto the bottom side of F-region along the magnetic field lines through electro-dynamical coupling during the late afternoon period (Röttger, 1981; Tsunoda, 2010; Huang et al., 2011; Retterer and Roddy, 2014; Tsunoda, 2015). Furthermore, an important activity at the equator, i.e., Pre-Reversal Enhancement (PRE), plays a significant role in influencing the plasma bubble growth and vertically lifting it after the sunset. PRE is a phenomenon that causes an enhancement in the zonal eastward electric field at the sunset terminator before the electric field reverses in the westward direction during the night (Abadi et al., 2015). This phenomenon creates a vertical electromagnetic (E x B) drift that influences the growth rate of the RTI by lifting the plasma to the height where the ion-neutral collision rate is low (Farley et al., 1970; Fejer and Kelley, 1980; Abadi et al., 2015). EPBs occur within hours right after sunset, and the degree to which it extends in the latitude and altitude depends solely on the magnitude of PRE (Farley et al., 1970; Abdu et al., 2003; Abadi et al., 2015).

The depletions in the equatorial plasma were initially identified from in-situ satellite measurements by Hanson and Sanatani (1973) and later confirmed by McClure et al. (1977). Since then, various techniques such as ground based observations (Woodman and La Hoz, 1976; Farley et al., 1970; Whalen, 1997; Kudeki and Bhattacharyya, 1999), airglow imagers (Sahai et al., 1994, 2000; Martinis and Mendillo, 2007), satellite-based in-situ measurements (Burke et al., 2004a; Park et al., 2005; Gentile et al., 2006; Stolle et al., 2006; Xiong et al., 2010; Dao et al., 2011) as well as Global Navigation Satellite Systems (GNSS) ground-based measurements (Basu et al., 1999; Carrano and Groves, 2007; Nishioka et al., 2008) have been used to study EPBs. Although these techniques contributed enormously towards the understanding of the ionospheric irregularities, they lacked in delivering critical information in one aspect or the other. For example, the ground-based sounders and GNSS ground receivers, despite that they provide crucial information related to the ionosphere and are globally distributed, remain restricted to a landmass. On the other hand, the in-situ satellite instruments explore the prevailing conditions in the ionosphere along the orbital track but fail to provide crucial insight into the vertical ionospheric conditions. Nonetheless, the GPS-RO technique, in recent times, has been widely used for ionospheric investigation owing to its extensive sounding capabilities along with high-resolution measurements; both globally as well as vertically for envisaging four-dimensional prospect of the ionosphere. (Wickert et al., 2001; Arras et al., 2008; Wickert et al., 2009; Carter et al., 2013; Liu et al., 2016; Tsai et al., 2017).

The GPS-RO is a space-based technique, which involves two satellites, i.e., GPS and Low Earth Orbiter (LEO), operating on a high-low satellite satellite tracking (HL-SST) mode (Wickert et al., 2001, 2009). The operational principle is mainly based on LEO satellites tracking the radio signals from the GPS satellites, causing the signal to bend as it penetrates the Earth's ionosphere and atmosphere. The fundamental observable, i.e., bending angle, obtained from bending of the signal at the point of closest approach to the Earth, is measured as an additional Doppler shift for accurate frequency and orbit geometry measurements (Kursinski et al., 1997, 1999). In the ionosphere, electron density profiles are obtained using the onion peeling algorithm (Lei et al., 2007). While, in the stratosphere and troposphere, temperature and pressure profiles are obtained using refractivity profiles (Wickert et al., 2002; Jakowski et al., 2004). In addition to providing such a wealth of information, this technique mitigates various technical shortcomings by operating under all weather conditions and providing long term stability without requiring calibration from time-to-time (Rocken et al., 1997). Due to GPS-LEO geometry, this technique provides measurements with a high vertical resolution that are globally distributed. In the past, various LEO missions contributed enormously towards radio occultation operations that led to the rise of one mission to another, starting from GPS/MET (GPS/METeorology), CHAMP (CHAllenging Minisatellite Payload), GRACE (GRAvity recovery and Climate Experiment), FormoSat-3/COSMIC (Formosa Satellite -3/Constellation Observing System for Meteorology, Ionosphere, and Climate) (Anthes et al., 2008; Wickert et al., 2009; Arras et al., 2010) to FormoSat-7/COSMIC 2 mission.

## 2 Data analysis

In this study, EPBs are analyzed using the GPS-RO measurements from the FormoSat-3/COSMIC satellites. The FormoSat-3/COSMIC mission is a constellation of six micro-satellites, which provide $\sim 2,000$ continuous real-time neutral atmospheric and ionospheric profiles daily (Anthes et al., 2008). However, after orbiting for more than 13 years and exceeding its planned lifespan of five years, the number of RO profiles have significantly reduced to approximately $20\%$ since the middle of 2016. This is because currently only one out of six satellites is operational under degraded mode (Chu et al., 2018). Nevertheless, this study comprises of measurements taken during the years 2007-2017 that includes nearly 5.5 million ionospheric profiles.

For investigating EPBs, *ionPhs* (*ion*ospheric excess *Pha*ses) data is used, which belongs to *level 1b* dataset. These FormoSat-3/COSMIC observation files are freely available on the web portal of COSMIC Data Analysis and Archival Center (CDAAC) database, which are managed by University Corporation for Atmospheric Research (UCAR), Colorado, United States of America. Furthermore, CDAAC also provides *'ScnLv1'* scintillation datasets, which contain off-line constructed S4 data calculated from $50\,\mathrm{Hz}$ that are recorded at $1\,\mathrm{Hz}$. But from the several thousand *ScnLv1* profiles that are retrieved daily, only less than one-fourth profiles can be reconstructed for the F-region altitude of the ionosphere (Tsai et al., 2017). Thus *ionPhs* datasets are retrieved and explored, which are almost five times more than the *ScnLv1*. The derivation of *ionPhs* profiles is based on the assumption of spherical symmetry; however, this is not valid for EPBs (Jakowski et al., 2004; Arras, 2010). These datasets are retrieved at $1\,\mathrm{Hz}$ sampling rate with $\sim 2\,\mathrm{km}$ of altitude resolution along the vertical range of $\sim 60\,\mathrm{km}$ above the Earth's surface up to the orbital height of the LEO ($\sim 800\,\mathrm{km}$).

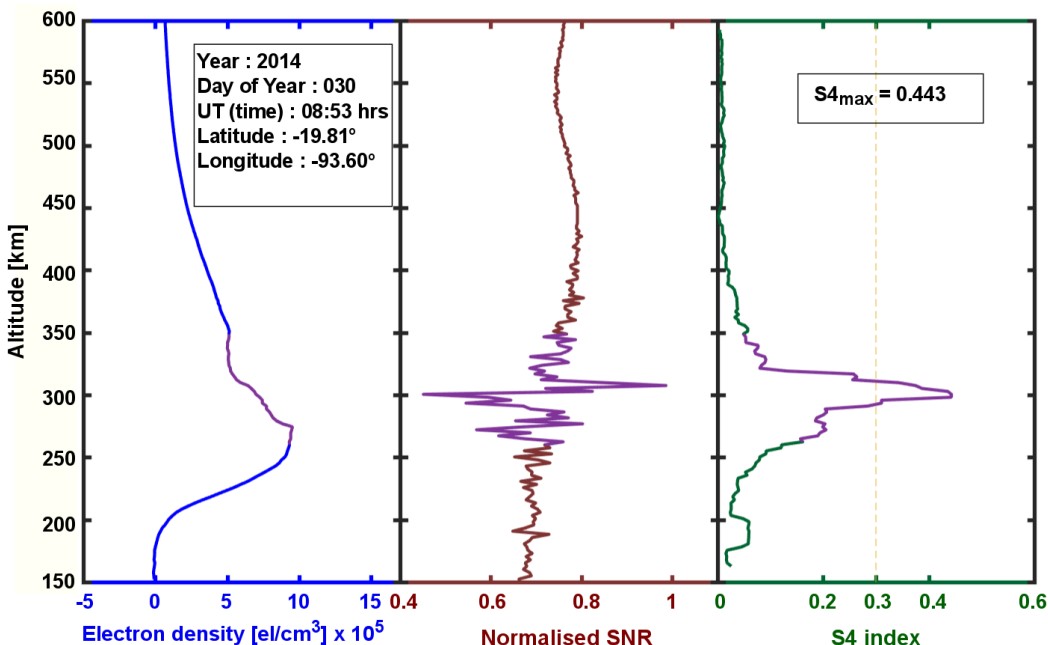

**Figure 1.** Electron density profile (ionPrf product) in conjunction with normalized SNR and derived S4 index (ionPhs product). The purple color line in the plot shows depletion in electron density and corresponding fluctuations of normalized SNR profile and high index values in the S4 plot.

In particular, raw Signal-to-Noise ratio (SNR) of the GPS L1 ($1,575$ MHz) *ionPhs* measurements are used. This is because the GPS L1 measurements show strong signal characteristics, and are received with a relatively higher intensity when compared to the GPS L2 ($1,227$ MHz) signals which are weaker and noisier. On the other aspect, SNR measurements are preferred over electron density profiles since they are straightway available, and no further treatment is required. Additionally, from literature, it is known that amplitude variation in the SNR profile has a direct influence on the vertical gradient of the electron density, which provides critical information on the underlying space weather conditions (Wickert et al., 2004; Arras et al., 2008). From Fig. 1, it is visible that the EPB's signature characterized by sharp depletion in the electron density corresponds to intense oscillations in the SNR profiles. Subsequently, these fluctuations produce a high value of amplitude scintillation index.

The scintillations caused by plasma bubbles are identified by deriving amplitude scintillation index, i.e., S4 index, from the SNR of the GPS L1 signals. This is because the variations in the SNR can be associated with the vertical changes in the electron density that mainly occur in line with the irregularities, e.g., EPBs (Hajj et al., 2002; Arras and Wickert, 2018). For subsequent analyses of the plasma bubbles, attributes of *ionPhs* datasets such as SNR of GPS L1 signal, Universal time, altitude,

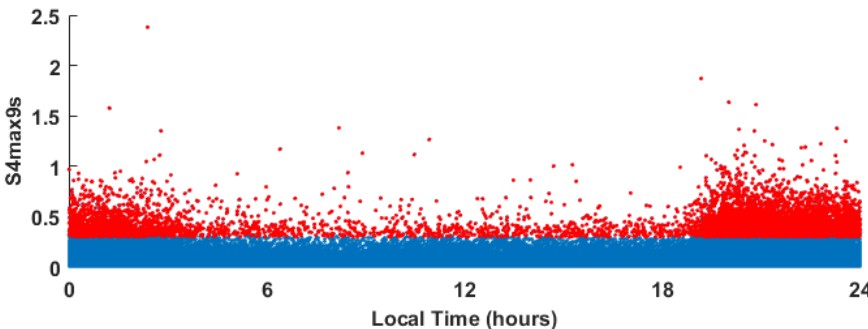

**Figure 2.** Plot of $S4max9sec$ as a function of local time (LT) during 2014. The blue dots represent the scintillation index less than 0.3, whereas strong scintillations are represented by the red dots having S4 index larger than 0.3.

latitude, and longitude are extracted. Eventually, the S4 index is computed from the raw SNR measurements, as described by Syndergaard (2006) in Eq. 1.

$$S4max9sec = \frac{\sqrt{\left\langle \left(I - \left\langle \bar{I} \right\rangle \right)^2 \right\rangle}}{\left\langle \bar{I} \right\rangle} \tag{1}$$

where $S4max9sec$ denotes the scintillation index calculated over nine seconds interval, $I$ is the square of the Signal-to-Noise
(SNR) ratio of L1 GPS signal, and the bracket $\langle \rangle$ stands for average taken over nine seconds. Also, a low pass filter is applied to the time series of nine seconds interval to obtain a new average of the intensity $\langle \bar{I} \rangle$ for constructing a long-term detrended $S4max9sec$ index (Syndergaard, 2006).

A simple representation of $S4max9sec$ versus local time during the year 2014 is depicted in Fig. 2, which shows scattered scintillation values caused due to varying electron density gradient. Additionally, it also highlights low $S4max9sec$ values
during the day and high values during the night. The high values observed during the night are due to the plasma instabilities in the F-region after sunset. Altogether about 0.5 million profiles were retrieved in 2014, out of which only 6,130 (i.e., $\sim 1.2\,\%$) global profiles were classified as strong scintillation events originating from possible plasma instabilities.

For this study, a scintillation event was classified based on the $S4max9sec$ index. Table 1 presents the different scintillation categories corresponding to different $S4max9sec$ (hereafter referred to as S4) index. Within this study, S4 index greater than
15 0.3 is quantified to be a strong scintillation event influenced by possible plasma bubbles. (Brahmanandam et al., 2012; Carter et al., 2013).

| S4 values | scintillation category | occurrence (2014) |
|---|---|---|
| $S4 \geq 1.0$ | high | 0.02% |
| $0.3 < S4 < 1$ | moderate | 1.19 % |
| $S4 \leq 0.3$ | low | 98.79% |

**Table 1.** Categorization of S4 index intensity.

## 3 Results

The FormoSat-3/COSMIC measurements between June 01, 2007, and December 31, 2017, were analyzed to understand the occurrence climatology of EPBs. This time-interval was selected to avoid the influences of orbit maneuvers in the data which were present until May 2007. Since the FormoSat-3/COSMIC satellites fly in non-sun-synchronous orbit, they effectively perform global soundings. However, to centralize this study in the equatorial region, only the measurements within the geographical latitudinal extent of $50°$ N/S are considered. By determining this limit, polar scintillation events are excluded, focusing explicitly on the equatorial ones. Also, the altitude range between $150\,km$ and $600\,km$ was specified to avoid the influences from the E-region and the noisier information from the GPS-RO profiles above $600\,km$.

### 3.1 Global distribution of EPBs

EPBs are field align irregularities, which occur along the geomagnetic equator and peaks during the time of year when the sunset terminator closely aligns with the magnetic field lines (Tsunoda, 1980, 1985). Fig. 3 reveals the global occurrence climatology of EPBs, covering a solar cycle, i.e., almost 11 years. The occurrence rate of EPBs is calculated as a ratio of a number of profiles that have S4 index greater than 0.3 to a number of all RO profiles within the specified grid integrated over the complete local time. Although the general occurrence of EPBs derived from the S4 index follows the course of the geomagnetic equator, the occurrence peak appears around and not directly at the geomagnetic equator. This result is expected because an equatorial anomaly reappears after the sunset, especially during high solar activity caused due to formation of an ionization trough at the magnetic equator (Aarons et al., 1981; Aarons, 1982). As a result, irregularities occur in the region of high plasma density, i.e., the crest of the equatorial anomaly. Previous results obtained using different techniques also showed strong scintillations in the crest latitudes compared with the dip equator (Basu et al., 1988, 2002).

Furthermore, the 11-year climatology outlines the descending-ascending-descending phase that corresponds to the solar cycle. Low occurrence rates were observed with the onset of the descending phase, until the solar minimum year 2009, with a peak in the South American sector. Whereas, during the ascending phase of the solar cycle, the occurrence rates increased until the solar maximum year 2014, with the peak stretch along the Atlantic-African region with each passing year. For the descending phase, after the solar maximum year 2014, the occurrence rates again deteriorate with the peak migrating towards the South American region. Throughout this climatology, a finite proportion, if not the peak occurrence, of EPBs were present in the South American region. One of the reasons conferred by Huang et al. (2001) suggests the existence of a weaker magnetic field in that region, which accounts for the RTI's irregularities, caused due to vertical plasma drift because of the zonal electric field during the sunset. On the contrary, Burke et al. (2004a) argued on the weak occurrence rates of EPB during high solar activity, citing reason towards increased E-region conductivity because of particle precipitation in the South Atlantic anomaly. Besides, McClure et al. (1998) proposed possible seeding from the gravity waves emerging from the troposphere in the Andes, which

has been investigated by Su et al. (2014). The author confirmed a good correlation only in the South American region due to gravity waves that originated in the intertropical convergence zone. However, in the Atlantic-African region, there was a positive but still weak correlation. For such correlations, the author referred that in addition to gravity waves, there subsisted other seed perturbations that produced plasma instabilities. From the annual EPB occurrence, almost negligible EPB occurrence was observed in the Atlantic-African, Asian, and Pacific regions during the low solar activity. Thus, some association of PRE to seed the EPBs in this region could be possible since the magnitude of PRE is principally affected by solar activity (Li et al., 2007; Stolle et al., 2008; Kil et al., 2009; Abadi et al., 2015). Therefore, a significant number of EPBs occur during high solar activity, when the magnitude of PRE is at its peak magnitude, while weak EPB occurrence rate is observed during low solar activity when PRE amplitude is also at its minimum.

## 3.2 Local time dependency

From the previous studies based on various probing techniques, it is evident that the EPBs are a night-time phenomenon, that includes small scale irregularities inside the bubble, which lead to turbulent structures that cause scintillations (Woodman and La Hoz, 1976; Whalen, 1997; Sahai et al., 2000; Gentile et al., 2006; Yokoyama, 2017). A general local time occurrence of EPBs during 2014 is presented in Fig. 4, which is based on the global soundings retrieved from the FormoSat-3/COSMIC satellites that fly in non-sun-synchronous orbit. The occurrence rate of EPBs, here, are based on the calculation similar to the global distribution occurrence, but for a different grid composition within the geographical latitudinal extent of $50°$ N/S. The rapid depletion of the E-region conductivity and the onset of PRE right after sunset, cause the plasma bubble to develop, i.e., 19:00 LT. This characteristic is noticeable from the local time occurrence of EPBs shown in 4 and agrees with the study carried out by Stolle et al. (2006) using CHAMP in-situ measurements. In general, a substantial occurrence of EPBs is observed during high solar activity year, while sparse EPBs are generated during low solar activity year. (Basu et al., 2002). In Fig. 5, a closer look at the occurrence of EPBs is presented, based on solar maximum (2014) and solar minimum (2009) year. The occurrence rate is calculated as a ratio of the S4 values greater than 0.3 to the total number of S4 profiles for a particular hour bin starting from 19:00 LT within the $50°$ N/S of the geographical latitudinal grid. From the bar plot, it is understood that EPBs culminate approximately one hour earlier, i.e., 21:00 LT, during solar maximum compared with the culmination time, i.e., 22:00 LT during the solar minimum year; which is in good agreement with the EPBs detected using CHAMP, and GRACE in-situ measurements by Xiong et al. (2010). However, local time characteristics manifested in this paper slightly differ from the local time distribution presented by Carter et al. (2013). In the author's paper, EPB's occurrence peaks about an hour later during the solar maximum year compared with the solar minimum year for all season-longitude. The local time occurrence characteristics presented in this paper agree well with the argument conferred by Burke et al. (2009) who suggests, that the slow process of gravity-driven currents over weak PRE magnitude influences the EPB occurrence to peak at a relatively later local time for the solar minimum year.

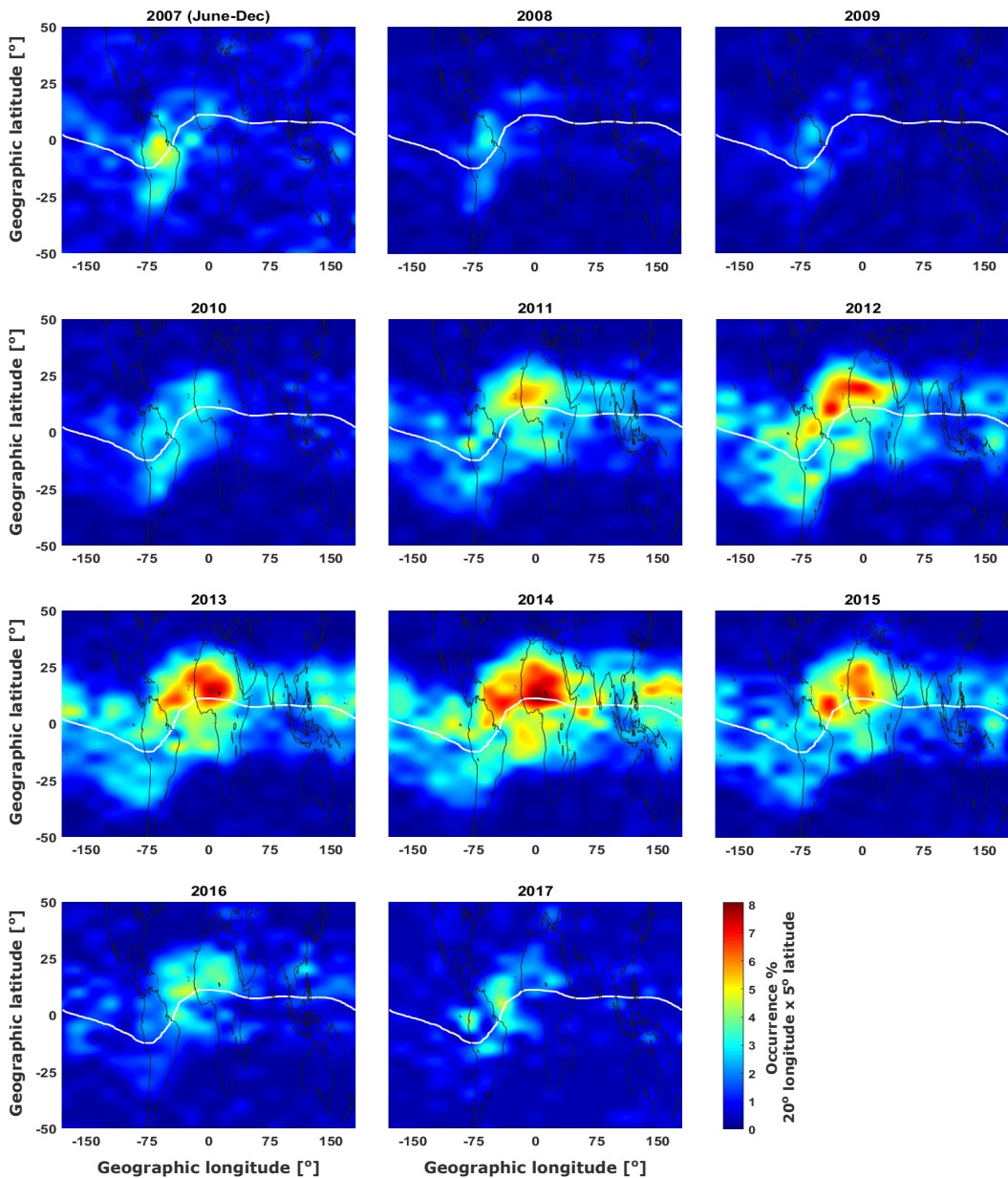

**Figure 3.** Plot of geographical latitude [°] v/s geographical longitude [°] of EPBs from mid-of 2007 to 2017. The white solid line depicts the geomagnetic equator.

### 3.2.1 Region-wise seasonal dependence of EPBs

Based on the argument put forth by Tsunoda (1985), the region-wise seasonal occurrence of plasma bubbles depends on the close alignment of the magnetic field line with the sunset terminator. In order to analyze the region-wise seasonal occurrence

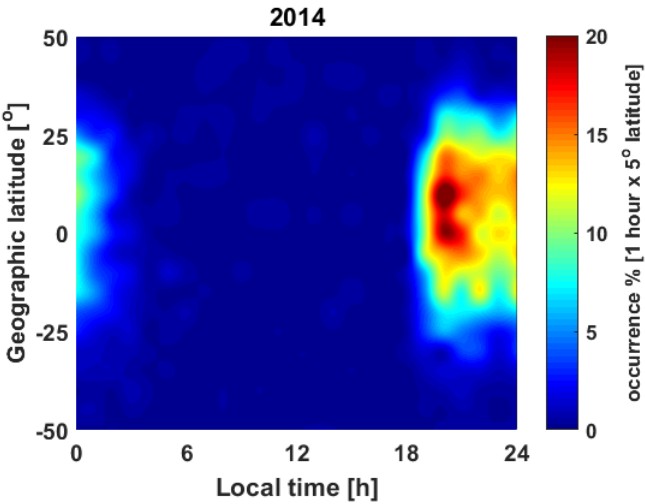

**Figure 4.** Latitudinal and local time dependence of equatorial plasma bubble occurrence during 2014.

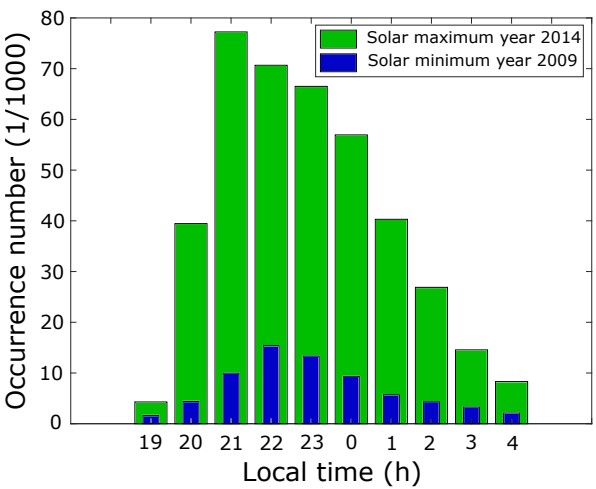

**Figure 5.** Occurrence of plasma bubbles based on local time during the solar minimum year (2009) and solar maximum year (2014) respectively.

characteristics of EPBs, the longitude extent was discretized in four different sectors of 90° each, which includes America (110°W-20°W), Africa (20°W-70°E), Asia (70°E-160°E) and Pacific (160°E-110°W). These longitude sectors are compared further with different seasons based on a three-month interval around each solstice and equinox. The region-wise seasonal occurrence envisaged in Fig. 6 is based on geomagnetic latitude with respect to local time, which is similar to the seasonal-longitude occurrence presented for solar minimum conditions (2007-2011) by Carter et al. (2013). In comparison, in this study, around 2.2 million profiles were analyzed to present EPB's distribution between 2012-2016 that covered the crest of the solar

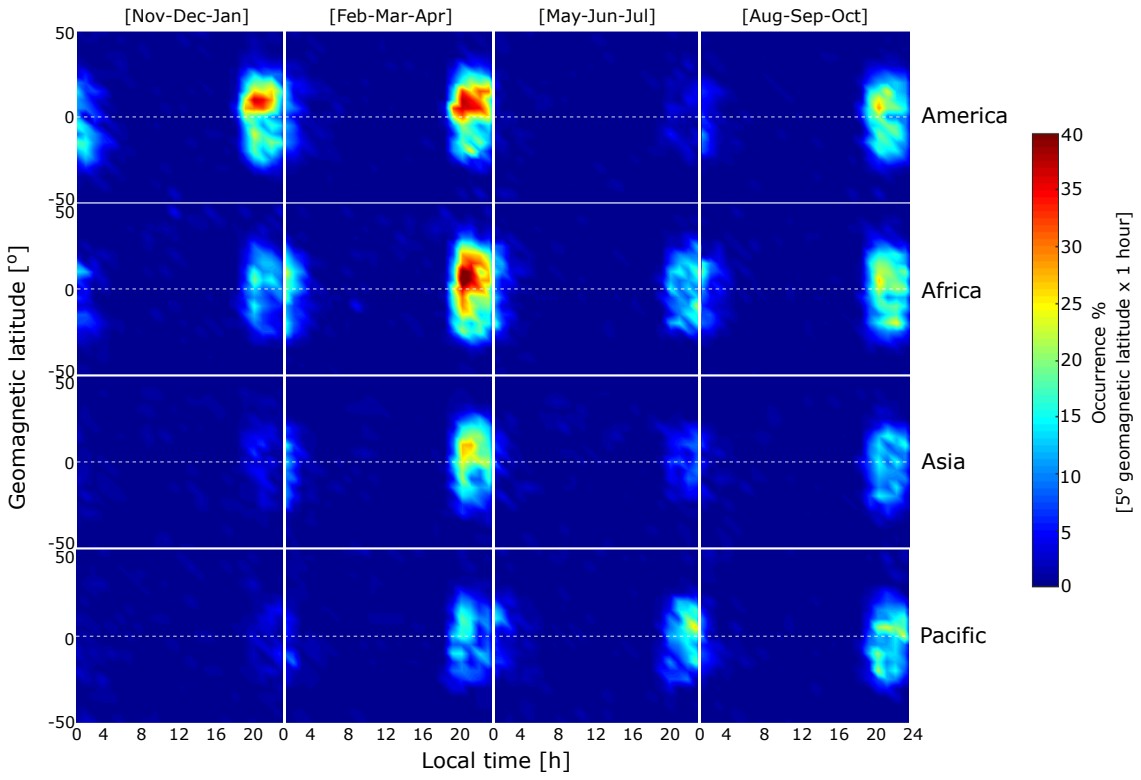

**Figure 6.** EPBs occurrence during the years 2012-2016 for different longitude sectors (regions) based on three-month intervals (season). White dashed lines represent geomagnetic dip equator.

cycle 24, i.e., 2014. In principle, EPBs are distributed on either side of the dip equator, with only one maximum on the positive side of the dip equator across all longitudes and seasons. On the contrary, two maxima on either side of the dip equator were observed by Carter et al. (2013) during solar minimum condition using FormoSat-3/COSMIC data, whereas only one peak at the dip equator was observed by Burke et al. (2004a) with Republic of China SATellite (ROCSAT)-1 observations in the period 2000-2002. The American region experienced a substantial occurrence of EPBs mostly across all seasons, except the June solstice (May-June-July), whereas, the African region encountered highest number of EPBs during the equinoxes and June solstice. Across all longitude sectors, Asia recorded the least occurrence rates of EPB for most of the seasons. In general, a maximum occurrence was observed during both the equinoxes in Africa and agree well with the results presented by Burke et al. (2004b) and Su et al. (2008), but it differs from the maximum equinoctial occurrence in America presented by Carter et al. (2013). The discrepancy observed could be due to measurements taken for two different solar conditions. Wherein, the American region experienced a peak occurrence of EPBs during the solar minimum conditions (Carter et al., 2013). However, during solar maximum conditions, the peak occurrence featured over the African region.

Furthermore, in the equinox and solstice seasons, asymmetries are observed, wherein, in the American region, almost negligible EPBs are detected during June solstice compared to the rest of the season. According to Tsunoda (1985), this was due to a vaster sunset time lag in the June solstice, which constraints the formation of EPBs. On the contrary, Africa, Asia, and the Pacific region recorded more EPBs during the June solstice compared to December solstice (November-December-January). But for this scenario, the sunset time lag approach could not justify the occurrence; however, it was rationalized by Nishioka et al. (2008) citing the reason for the integrated flux tube conductivities in the F-region and its seasonal occurrence, which proved to be favorable for the solstice asymmetry in Africa, Asia, and Pacific sectors. For the equinox asymmetry, America, Africa, and Asia encountered a significant occurrence in March equinox (February-March-April) compared to September equinox (August-September-October), except for the Pacific region, which agrees well with Burke et al. (2004b). In general, the Eastern hemisphere, e.g., Asian and parts of Pacific sectors, recorded few EPBs, because of the dominant magnetic field at the equator. Whereas comparably more EPBs were observed in the region of a relatively weak equatorial magnetic field, i.e., at the American and African longitudes (Burke et al., 2004a, b).

## 3.3 Altitude variations and solar cycle dependency

The FormoSat-3/COSMIC measurements provide height dependent information, which is valuable as compared to the measurements obtained from the other contemporary techniques for investigating plasma bubbles on a global scale. Based on the generalized notion, the EPBs are generated in the bottom side of the F-region as a consequence of the RTI, and move upwards through the electrodynamic process (Whalen, 1997; Kelley, 2009; Woodman, 2009). Fig. 7 shows the altitude distribution of EPBs on an annual basis and manifests that the occurrence of plasma bubbles is dependent on different conditions of the solar activity. The study also revealed that the periodic variation in the solar cycle plays an indirect role in influencing the vertical occurrence range of the plasma bubbles. Thus, during high solar activity, i.e., 2014, EPBs were spread over a sizeable range, while during low solar activity, i.e., 2009, a smaller altitude range was covered. Besides, the occurrence peak of EPBs during 2014 was at an altitude of $\sim 420\,\text{km}$, while during 2009, it occurred around $\sim 240\,\text{km}$. The altitudinal uplift of EPBs is mainly due to the magnitude of PRE, which is dependent on the solar activity. (Fejer et al., 1999; Stolle et al., 2008; Abadi et al., 2015; Liu et al., 2016). In addition, EPBs primarily generated at the geomagnetic equator elongates in latitude due to the dominance of PRE (Abdu et al., 2003; Anderson et al., 2004; Liu et al., 2016). This is obvious in the altitude distribution of the plasma bubbles, wherein during low solar activity, EPBs were almost contained at the geomagnetic equator, while during high solar activity, EPBs were spread out on either side of dip equator Liu et al. (2016).The growth rate and the altitudinal variation of EPBs were an outcome of degenerated conductivity in the E-region along with an enhanced zonal electric field at the sunset (Farley et al., 1970; Stolle et al., 2008; Su et al., 2014). Ideally, PRE lifts the plasma in the F-layer by means of E×B drift to an altitude where the neutral-ion collision frequency is low, which is inversely proportional to the growth rate of plasma bubble (Fejer et al., 1999; Abadi et al., 2015). In the process, EPBs continue to proceed higher in altitude until the eastward electric field on the top of the bubble becomes zero, which eventually causes them to decay (Krall et al., 2010).

From the occurrence climatology presented in this paper, it is apparent that the influence of PRE causes EPBs to materialize in accordance with the solar activity. Thus, more EPBs are detected during maximum solar activity compared to minimum (Basu

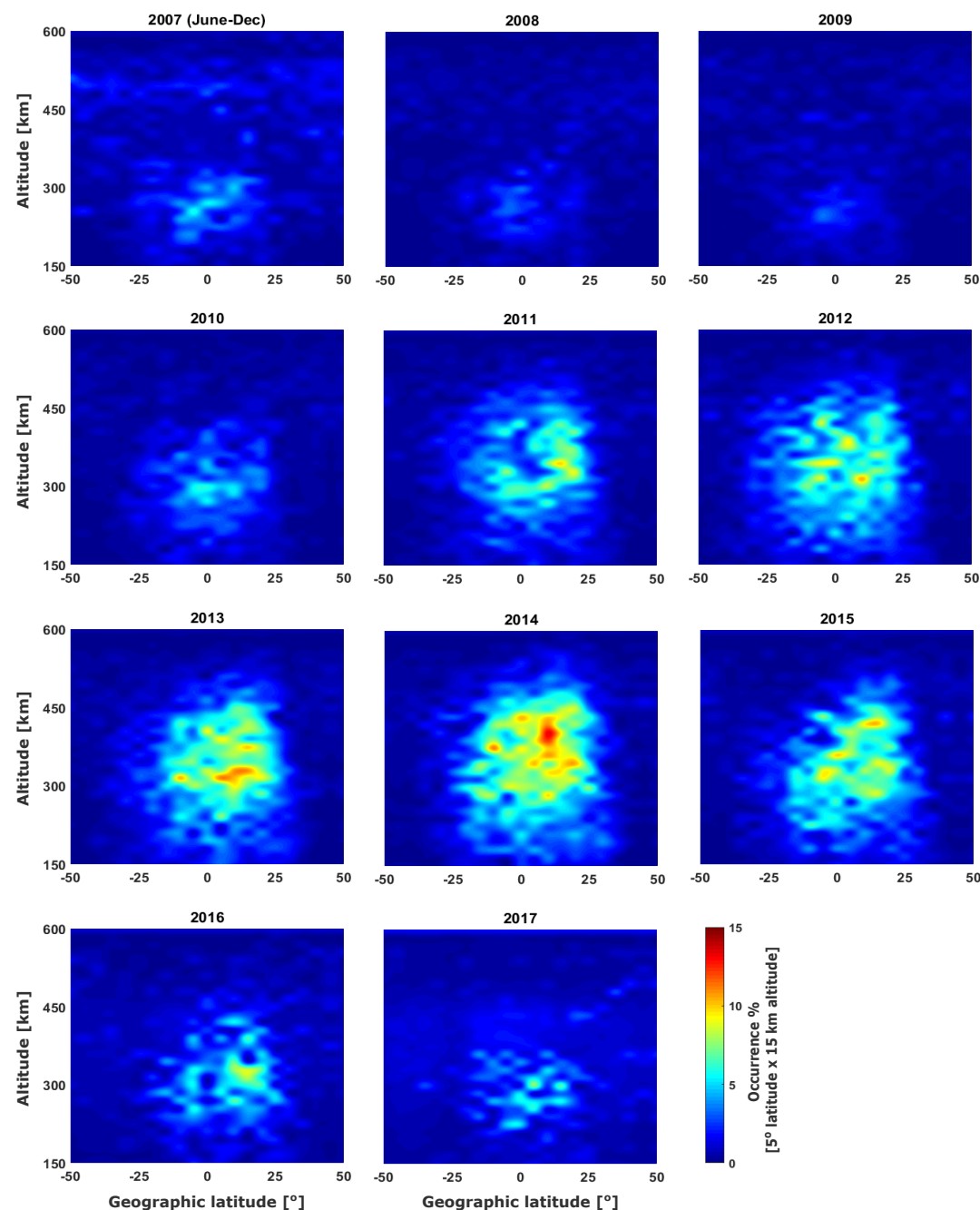

**Figure 7.** Plot of geographical latitude v/s altitude of equatorial plasma bubbles for showing vertical distribution during the years between mid-of 2007 and 2017.

et al., 2002). A brief analogy in support of the argument is presented in Fig. 8, which shows the sunspot cycle and relative

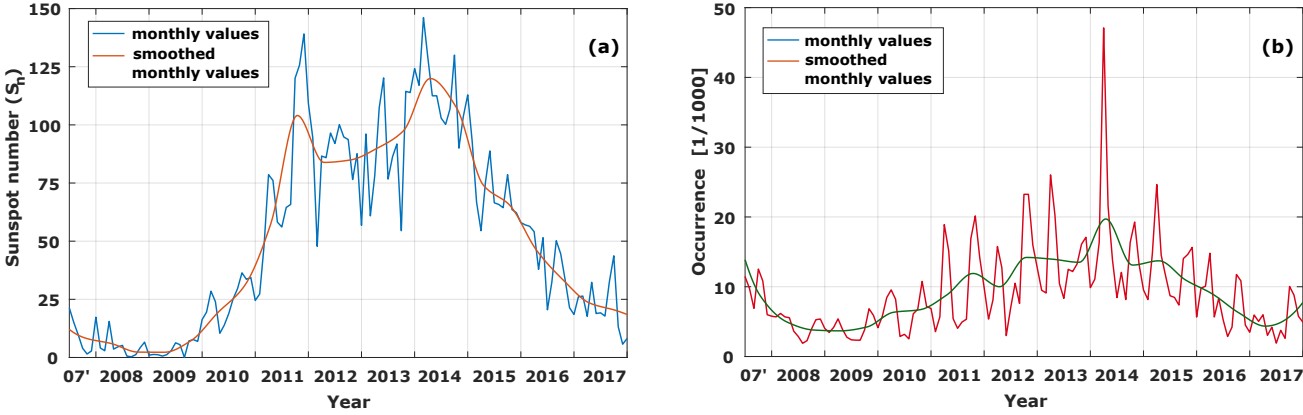

**Figure 8.** Comparison plot of (a) Sunspot cycle (b) Occurrence trend of equatorial plasma bubbles from mid-of 2007 to 2017, having monthly values and smoothed monthly values using low pass filter.

occurrence numbers of EPBs with semi-annual structures across different years. Further, Fig. 8a depicts the current sunspot cycle represented by the monthly sunspot numbers(blue solid line) and a smoothed curve (orange solid line), whereas Fig. 8b shows an annual occurrence trend of plasma bubbles characterized by monthly (red solid line) and smoothed monthly values (green solid line) from mid-of 2007 to 2017. On the global spectrum, the EPBs occur in line with solar activity; however, it is not a typical scenario on a regional basis. Nishioka et al. (2008) showed that the dependence of solar activity in specific longitude sectors does not influence the occurrence rate of EPBs. For example, EPBs in the African and Asian sectors appear in congruence with the solar cycle; however, the same is not observed in the American sector, as revealed in Fig. 3. This most likely could be due to the presence of gravity wave perturbations, which seed EPBs despite weak PRE magnitudes during solar minimum conditions in the South American region (Burke et al., 2004a; Stolle et al., 2008; Su et al., 2014).

## 4 Conclusions

This paper provides a brief occurrence climatology of EPBs covering around 10.5 years of GPS-RO measurements derived from FormoSat-3/COSMIC. The scintillations induced in the radio waves caused by the EPBs were detected using an amplitude scintillation index known as the S4 index. By classifying the S4 data, subsequent analyses are carried out by exploiting the strong scintillation events. In this study, EPBs occur at the crest anomaly latitudes along the geomagnetic equator and have peak occurrence oscillating between America and Africa for solar minimum and solar maximum years, respectively. Furthermore, the annual global distribution of EPBs showed good congruency with solar activity, especially in Africa. Thus implying on the influence of vertical drift from PRE, which also depends on the solar activity. However, there is no apparent dependence on the solar cycle in the American sector. In hindsight, gravity-driven currents are known to have a good correlation on the occurrence of plasma bubbles solely in the American area. Therefore, it is presupposed that the EPBs are triggered with different seed

perturbations for different regions. From the local time occurrence, EPBs are apprehended to develop post-sunset around 19:00 LT, right after the enhancement in the zonal eastward electric field at the sunset. Moreover, EPBs generated during solar maximum year peaks at an hour earlier compared to EPBs during the solar minimum year. This implicates a dependency on PRE, which has a larger magnitude of vertical plasma drift during high solar activity compared with low solar activity. On the other hand, region-wise seasonal occurrence shows maximum EPBs in Africa during March equinox. Almost, in all longitude sectors, more EPBs were detected in the March equinox compared to September equinox. Whereas for solstice months, it agrees with the argument from Tsunoda (1985), wherein more EPBs were encountered at longitudes with positive (negative) declination during June (December) solstice and have good agreement with Burke et al. (2004b), Su et al. (2008), and Carter et al. (2013). Recently, Xiong et al. (2010) articulated based on a comparative study of EPBs using CHAMP and GRACE in-situ measurements, that more EPBs get detected at an altitude below $300\,km$, compared to the above. However, since the in-situ measurements encounter EPBs at an orbit altitude usually above $\sim 400\,km$, only some signatures of EPBs, e.g., only small dips in the plasma density, are just detected. Thus, the GPS-RO endorses to be a convenient tool for investigating the EPBs because of their vertical soundings at the same time provide global resolution. Meanwhile, these EPBs, which are provoked by PRE, show a strong dependence on the periodic variation in solar activity with a greater altitude extent during high solar activity. In principle, throughout the global analyses, a comparison with the sunspot cycle with the annual EPB occurrence reveals a strong dependence on solar activity. Overall, the GPS-RO technique seems promising in understanding the global EPBs and can also perform as a complementary technique in analyzing such ionospheric irregularities because of unique measurements available as a result of vertical scans.

*Code availability.* TEXT

*Data availability.* Ionospheric radio occultation data is based on FormoSat-3/COSMIC satellite mission available from CDAAC (http://www.cosmic.ucar.edu). The dataset for the solar sunspot number is obtained from Sunspot Index and Long term Solar Observations website (http://www.sidc.be/silso/datafiles)

**Appendix A**

**A1**

*Author contributions.* A. Kepkar performed the analysis and drafted the manuscript with the help of C. Arras and J. Wickert. H. Schuh, M. Alizadeh and L.C. Tsai provided with constructive scientific advices.

*Competing interests.* The authors declare that they have no conflict of interest.

*Disclaimer.* TEXT

*Acknowledgements.* The authors recognize the efforts of FormoSat-3/COSMIC team and are grateful for providing the measurements. C. Arras acknowledges the support from Deutsche Forschungsgemeinschaft (DFG) Priority Program DynamicEarth SPP1788. A. Kepkar acknowledges support from DFG under SCHU 1103/15-1.

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
