# Peer review of "Occurrence climatology of equatorial plasma bubbles derived using FormoSat-3/COSMIC GPS radio occultation data"

_Annales Geophysicae, 2019_

## Referee Comment (RC1) · Anonymous Referee #1 · 9 Apr 2019

This paper investigated the distribution of bubbles as a function of longitude, latitude, altitude, local time, and year (solar cycle) using the FormoSat-3/COSMIC radio occultation data acquired in 2008-2016. Their results show an agreement with the occurrence climatology of bubbles derived by other observations.

On the scientific aspect, this paper does not deliver any new findings regarding bubbles; the behavior of bubbles (dependence on the geographic and geophysical parameters) is well established, even if this paper does not report. However, the paper demonstrates that GPS RO signal can be a good proxy for the detection of bubbles. The results are acceptable, but writing is so bad, so I recommend resubmission of the paper.

For most sentences, I could not progress to the next sentence without pointing out a problem. Below are some examples.

**Title**

May change "Occurrence climatology of equatorial plasma bubbles derived using the FormoSat-3/COSMIC GPS radio occultation data"

**Abstract**

The whole sentences should be revised.

Line 2-3: The words "emerging", "prominent" are not necessary.

Line 4-5: "For investigating the plasma bubbles, a nine-year (2008-2016) long time series of signal-to-noise ratio data are used from the vertical GPS radio occultation profiles." This is bad structure. I would write "The occurrence climatology of bubbles is derived using the vertical GPS radio occultation data in 2008-2016 by the FormoSat-3/COSMIC mission."

Line 8-9: "Dependence on the solar cycle as well as distinctive seasonal variation is observed when analyzed for different years." -> The distribution of bubbles shows the dependence on season, longitude, and solar cycle"

The words "depreciated" and "personifies" do not sound good expressions.

**Sections 1-3**

**There are many awkward expressions. Too much work to point out all of them**

**Conclusions**

Page 10 Line 28: " a nine-year comprehensive study of equatorial plasma bubbles …" It sounds that the authors have studied bubbles for nine years.

Page 10 Line 32: There is no "striking" finding of this study. The solar cycle dependence of the bubble activity is already very well known, and this study just has identified the known phenomenon using the RO data.

Page 10 Line 33-Page 11 Line 1: The concentration of bubbles near the magnetic equator is already well known fact, and it is not an intriguing characteristic at all.

Page 11 Line 1: "The rapid depletion of E-layer post sunset cause…" -> The rapid plasma loss in the E layer after sunset causes …

Page 12 Line 1: "The study reveals the influence of solar cycle, which facilitates the contraction and expansion of plasma bubbles across the complete altitude range." Does the solar cycle contract or expand bubbles? What does this mean?

---

## Referee Comment (RC2) · Anonymous Referee #2 · 16 Apr 2019

The present work describes characteristics of occurrence of equatorial plasma bubbles as a function of longitude, local time, altitude and solar cycle, using COSMIC radio occultation data. The year to year variations of the longitudinal distribution of S4 index (Figure 3), Longitudinal difference of the seasonal variation of occurrence of S4 (Figure 6), and solar cycle dependency of maximum altitude of S4 (Figure 7) are new results and important contribution for scientific community.

Therefore I would recommend that the present work could be published in Angeo.

Minor comments are to be considered by authors:

Page 3 line 27, "L2 signal is close to the critical frequency of the ionosphere„": The

authors will need to explain why they mention it. The frequency of L1 and L2 are 1.575 GHz and 1.227 GHz, respectively, are very close to each other, and these are far from the critical frequency of the ionosphere as far as I understand. If they have a special reason to not using L2, please explain it more detail.

Page 4, line 4, "S4max9sec denotes „„ 9 seconds interval": Please explain why they used 9 seconds to calculate S4, instead of the original S4 value of one second. Page 4, line 5-6, "A low pass filter is applied to the time series of these values„„": What time series ?, of 9 seconds interval ?

Page 6 line 9, "S4 index is derived „„ understanding the occurrence of plasma bubbles": The authors interpreted the observed S4 larger than 0.3 are all caused by plasma bubbles. The presence of scintillation (or spread F), however, does not mean that it is due to Plasma Bubbles. There are possibilities of other sources such as ionospheric waves (TID, MSTID). The authors could comment on it.

Page 9, line 3 "In order to have in detail „„ analysis was performed" which is shown in Figure 6 ?

Page 11, line 5 "in the African sector during June solstice": Isn't it March Equinox ? (see Figure 6).

END of Report

---

## Referee Comment (RC3) · Anonymous Referee #3 · 25 Apr 2019

Global Climatology of Equatorial Plasma Bubbles based on GPS Radio Occultation from FormoSat-3/COSMIC by Kepkar et al looks at the seasonal, longitudinal, annual, altitude and local time variations in the occurrence of equatorial plasma bubbles as indicated by the COSMIC S4 index. The authors have failed to highlight the novelty of the work, and as such I can not recommend it for publication. Their work is very similar to Carter et al 2013, whom they cite in reference to variation with solar activity. However, they have failed to discuss the work in the context of this paper even though Carter et al 2013 looked at the seasonal, longitudinal, annual and local time variations of equatorial plasma bubbles as indicated by the COSMIC S4 index. Indeed Figure 6 of this paper is very similar to Figure 4 of Carter et al 2013. The years of data used are

different. However, the differences and similarities between Figure 6 and the results from Carter et al 2013 are not discussed and it is showing how this work provides something new that is currently missing from the manuscript.

The main findings chosen to be highlighted in the abstract are "The analysis revealed that the F-region irregularities, associated with plasma bubbles occur mainly post sunset close to Earth's geomagnetic equator. Dependence on the solar cycle as well as distinctive seasonal variation is observed when analyzed for different years. In contrast to the other ionospheric remote sensing methods, GPS Radio Occultation technique uniquely personifies the activity of the plasma bubbles based on altitude resolution on a global scale." Taking each sentence in turn below it can be seen that no new information is currently being highlighted by the authors "The analysis revealed that the F-region irregularities, associated with plasma bubbles occur mainly post sunset close to Earth's geomagnetic equator."- This has been known and written in many papers, including ones such as Sultan 1996 and others that look at the mechanism and growth rate of post sunset plasma bubbles. "Dependence on the solar cycle as well as distinctive seasonal variation is observed when analyzed for different years." -Bourke et al 2004 looked at the climatology of plasma bubbles for both low and high solar activity, and Carter et al 2013 looked at the climatology of plasma bubbles using COSMIC S4 as a function of year "In contrast to the other ionospheric remote sensing methods, GPS Radio Occultation technique uniquely personifies the activity of the plasma bubbles based on altitude resolution on a global scale." –As mentioned throughout this review, Carter et al 2013 used COSMIC RO data to look at plasma bubbles, so more is needed to make this a new finding.

Minor comments: In section 2.1.1 Derivation of amplitude scintillation index it is unclear if the authors have used the provided s4max9sec data and are explaining how it is derived, or if they have used raw data and re-analysed it themselves. If it is the later the reasoning also needs to be made clear to the reader.

It is unclear why the authors have chosen to use the average S4 rather than some

occurrence calculation. Averages can be misleading if the distribution between cells varies, or the number of points vary etc. A justification should be added or another way to demonstrate the data should be used.

In Figures 5 & 6 the captions state that the Figures show the EPB occurrence. In Figure 5 the numbers seem very low for this to be the case and in Figure 6 it is clearly the S4 average again, this is inconsistent, confusing and needs to be fixed

In order to make this work worthy of publication the authors need to carefully discuss the results in the context of similar papers that are currently not included in the discussion of the results. Following this they need to assess and highlight what is new and different to determine if the work is novel.
* * *

---

## Author Comment (AC1) · 28 May 2019

Anonymous reviewer #1

This paper investigated the distribution of bubbles as a function of longitude, latitude, altitude, local time, and year (solar cycle) using the FormoSat-3/COSMIC radio occultation data acquired in 2008-2016. Their results show an agreement with the occurrence climatology of bubbles derived by other observations.

On the scientific aspect, this paper does not deliver any new findings regarding bubbles; the behaviour of bubbles (dependence on the geographic and geophysical parameters) is well established, even if this paper does not report. However, the paper demonstrates that GPS RO signal can be a good proxy for the detection of bubbles. The results are acceptable, but writing is so bad, so I recommend resubmission of the paper.

For most sentences, I could not progress to the next sentence without pointing out a problem. Below are some examples.

➔ We like to thank reviewer #1 for taking the time to read and comment on our paper. We are grateful for the valuable comments which will help us to improve our paper distinctly. Regarding the writing of the paper, we have taken your comments into consideration and made appropriate modifications.

Title

May change "Occurrence climatology of equatorial plasma bubbles derived using the FormoSat3/COSMIC GPS radio occultation data"

➔ Thank you for suggesting the title. We will replace the existing title with the suggested one, as the latter is more appropriate with the topics discussed in the paper.

Abstract

We are grateful for the constructive inputs for improving the abstract and will consider all of them in the revised manuscript.

The whole sentences should be revised.

➔ Rewrote the abstract as follows:
"The GPS Radio Occultation technique is used to detect the equatorial plasma bubbles in the F-region, which are characterized by depleted regions of electron density. The occurrence climatology of bubbles is derived using the vertical GPS radio occultation data in 2008-2016 by the FormoSat-3/COSMIC mission. The plasma bubbles are identified based on the S4 index, derived from the signal-to-noise ratio of the radio occultation profile. The analyses revealed that, the F-region irregularities associated with plasma bubbles occur mainly post sunset close to Earth's geomagnetic equator. The distribution of bubbles shows the dependence on season, longitude, and solar cycle. Through this paper, a modest attempt is made to show that GPS Radio Occultation can be used as complementary technique to investigate plasma bubbles. Additionally, the advantage in using radio occultation data is

that, we do not only get information on the occurrence of the equatorial bubble events but also on its altitude distribution."

Line 2-3: The words "emerging", "prominent" are not necessary.

➔ The words "emerging", "prominent" are removed in the revised abstract as suggested and agree that it was simply unnecessary.

Line 4-5: "For investigating the plasma bubbles, a nine-year (2008-2016) long time series of signal-to-noise ratio data are used from the vertical GPS radio occultation profiles." This is bad structure. I would write "The occurrence climatology of bubbles is derived using the vertical GPS radio occultation data in 2008-2016 by the FormoSat-3/COSMIC mission."

➔ Thank you very much for the ideal structure of the sentence. The existing sentence is replaced with the suggested sentence in the revised abstract.

Line 8-9: "Dependence on the solar cycle as well as distinctive seasonal variation is observed when analyzed for different years." -> The distribution of bubbles shows the dependence on season, longitude, and solar cycle.

➔ According to reviewer suggestion, the changes are incorporated in the revised abstract and we agree that it improves the quality of the sentence

" The words "depreciated" and "personifies" do not sound good expressions.

➔ The words "depreciated" and "personifies" are removed from the abstract and agree that inclusion of such words did not make expressions sound good.

Sections 1-3 There are many awkward expressions. Too much work to point out all of them.

➔ Thank you very much for going through the section in-detail. We agree that sections 1-3 are not well written and therefore we will rephrase it to avoid awkward expressions to the best of our ability in our revised manuscript.

Conclusions

➔ We are grateful to reviewer #1 for his in-detail review on the conclusion of the paper and included it in our revised manuscript in line with your suggestion.

Page 10 Line 28: " a nine-year comprehensive study of equatorial plasma bubbles …" It sounds that the authors have studied bubbles for nine years.

➔ Thank you for providing the perspective to the sentence. We have rewritten and made changes to the phrase as: "In this paper, a nine-year global climatology of EPBs is presented using GPS-RO measurements obtained from the FormoSat-3/COSMIC mission."

Page 10 Line 32: There is no "striking" finding of this study. The solar cycle dependence of the bubble activity is already very well known, and this study just has identified the known phenomenon using the RO data.

→ According to the reviewer suggestion, we agree that solar cycle dependence of the bubble activity is already well known. However, we tried to show it from the perspective of RO data and agree the unnecessity of the word 'striking'. Therefore, the word is removed from the manuscript according to authors suggestion.

Page 10 Line 33-Page 11 Line 1: The concentration of bubbles near the magnetic equator is already well known fact, and it is not an intriguing characteristic at all.

→ Thanks a lot for this hint. Yes, the bubble climatology is already well known from other measurement techniques. In this paper we like to show the time statistical distribution of this phenomenon based on GPS RO data. We added some sentences comparing our results with those from former publications (Carter et al.,2013, Liu et al.,2016) and also with publications that are not based on GPS RO data like (Stolle et al.,2006, Xiong et al.,2010). The main finding portrays the shifting of the peak EPB occurrence from South American to African sector along geomagnetic equator as we proceed from solar minimum to solar maximum and we intend to include this in the revised manuscript.

Page 11 Line 1: "The rapid depletion of E-layer post sunset cause…" -> The rapid plasma loss in the E layer after sunset causes …

→ We are thankful for different formulation of the sentence which is more crisp and clear. Therefore, changes are done in line with the reviewer suggestion.

Page 12 Line 1: "The study reveals the influence of solar cycle, which facilitates the contraction and expansion of plasma bubbles across the complete altitude range." Does the solar cycle contract or expand bubbles? What does this mean?

→ Thank you for providing your perspective on this sentence. The study falls in line with the different conditions of solar activity, wherein during solar minima the scintillation activity contracts, whereas as we proceed the solar maxima, the scintillation activity expands along the altitude range. Taking your view into consideration the rephrased sentence is:" The study reveals that the periodic variation in the solar cycle has an indirect role in the vertical occurrence range of the plasma bubbles, that covers a large range during solar maximum and lower altitude range during solar minimum condition.

→ Finally, we would like to appreciate reviewer #1 for his time in reviewing this paper comprehensively. We have included the suggestions and the changes which will significantly improve the quality of the paper.

References:

- Carter, B. A., Zhang, K., Norman, R., Kumar, V. V., & Kumar, S. (2013). On the occurrence of equatorial F-region irregularities during solar minimum using radio occultation measurements. Journal of Geophysical Research: Space Physics, 118(2), 892-904.
- Liu, J. Y., Chen, S. P., Yeh, W. H., Tsai, H. F., & Rajesh, P. K. (2016). Worst-case GPS scintillations on the ground estimated from radio occultation observations of FORMOSAT-3/COSMIC during 2007–2014. Surveys in Geophysics, 37(4), 791-809.
- Stolle, C., Lühr, H., Rother, M., & Balasis, G. (2006). Magnetic signatures of equatorial spread F as observed by the CHAMP satellite. Journal of Geophysical Research: Space Physics, 111(A2).
- Xiong, C., Park, J., Lühr, H., Stolle, C., & Ma, S. Y. (2010). Comparing plasma bubble occurrence rates at CHAMP and GRACE altitudes during high and low solar activity. Annales Geophysicae, 28(9), 1647-1658.

---

## Author Comment (AC2) · 28 May 2019

Anonymous reviewer #2

The present work describes characteristics of occurrence of equatorial plasma bubbles as a function of longitude, local time, altitude and solar cycle, using COSMIC radio occultation data. The year to year variations of the longitudinal distribution of S4 index (Figure 3), Longitudinal difference of the seasonal variation of occurrence of S4 (Figure 6), and solar cycle dependency of maximum altitude of S4 (Figure 7) are new results and important contribution for scientific community.

 Therefore, I would recommend that the present work could be published in Angeo.

Minor comments are to be considered by authors:

➔ We would like to thank the reviewer for his valuable comments and his perspective inputs for the significant improvement of the paper and immensely grateful for the encouragement on the manuscript. We have addressed all the minor comments that are put forth by the reviewer.

Page 3 line 27, "L2 signal is close to the critical frequency of the ionosphere,,,": The authors will need to explain why they mention it. The frequency of L1 and L2 are 1.575 GHz and 1.227 GHz, respectively, are very close to each other, and these are far from the critical frequency of the ionosphere as far as I understand. If they have a special reason to not using L2, please explain it more detail.

➔ The L1 and L2 are close to each other. But L2 signal is relatively closer to the critical frequency of the ionosphere and is therefore more affected. Unfortunately, the L2 signal is much weaker and nosier to compared to L1 signal and therefore it is not always possible to distinguish between information and noise.  Hence L1 signal is used for the occurrence climatology of bubbles.

Page 4, line 4, "S4max9sec denotes ,,, 9 seconds interval": Please explain why they used 9 seconds to calculate S4, instead of the original S4 value of one second.

➔ We use raw 1Hz measurements from FormoSat-3/COSMIC, in which we get observations per one second. In order to compute S4 index, a running average is required and 9 second approximation proved to be favourable as seen from previous studies (Carter et al. (2013), Tsai et al. (2017)). The S4 value of one second which is mentioned by the reviewer is not obtained by raw 1Hz measurement, but they are retrieved from 50 Hz measurements recorded at 1 Hz (Syndergaard, 2006). Since we receive reduced number of 50 Hz data in the F-region (i.e. by factor of 5) when compared to raw 1 Hz measurement, we exploit and compute S4 index from the raw 1 Hz measurements.

Page 4, line 5-6, "A low pass filter is applied to the time series of these values,,,": What time series? of 9 seconds interval?

➔ Thank you for your perspective comments. The time series refers to the average of 9 seconds interval according to the calculation of S4 index published by Syndergaard, 2006.

Page 6 line 9, "S4 index is derived ,,, understanding the occurrence of plasma bubbles": The authors interpreted the observed S4 larger than 0.3 are all caused by plasma bubbles. The presence of

scintillation (or spread F), however, does not mean that it is due to Plasma Bubbles. There are possibilities of other sources such as ionospheric waves (TID, MSTID). The authors could comment on it.

➔ We completely agree with the statement made by the reviewer regarding scintillation. Spread F is a more general term for plasma bubble and other ionospheric waves. However, through literature it has been known that the ionospheric waves originate at mid latitudes and therefore also called as "midlatitude spread F" (Kelley, M. C., and Miller, C. A. (1997)). Since we use SNR based RO data, which are sensitive to strong vertical changes in the electron density, we assume that it is unlikely to see a TID signature in SNR; since TIDs have long vertical wavelength of more than 100 km, we do not expect that they are able to compress ions into compact layers. In addition, some of the previous studies based on Europe and South America (Brazil) conducted by Otsuka et al. (2013) and Figueiredo et al. (2018) respectively show that, the MSTIDs mostly occur in winter and during day time. However, very low occurrence percentage of nighttime MSTIDs reported by Figueiredo et al. (2018) near equator could be neglected, while keeping in mind the relative local time occurrence of EPBs.

Page 9, line 3 "In order to have in detail „„ analysis was performed" which is shown in Figure 6?

➔ Thank you for pointing out this. The statement made corresponds to the Figure 6. We agree that there is not much detail explanation done contributing to the figure and therefore we plan to explain it in more detail. However as of now, the sentence" In order…" will be removed in the revised manuscript and replaced with a modified phrase.

Page 11, line 5 "in the African sector during June solstice": Isn't it March Equinox? (see Figure 6).

➔ Thank you for bringing this to our notice. It is actually the March Equinox, where in the scintillation activity is dominant when compared to rest of the seasons.

Finally, we would like to appreciate the reviewer for his time in reviewing this paper extensively and have considered all his comments in order to further improve the quality of the paper.

Reference:

- Carter, B. A., Zhang, K., Norman, R., Kumar, V. V., and Kumar, S (2013), On the occurrence of equatorial F-region irregularities during solar minimum using radio occultation measurements, Journal of Geophys. Res., 118, 892–904, https://doi.org/10.1002/jgra.50089.
- Figueiredo, C. A. O. B., Takahashi, H., Wrasse, C. M., Otsuka, Y., Shiokawa, K., & Barros, D. (2018). Medium-Scale Traveling Ionospheric Disturbances Observed by Detrended Total Electron Content Maps Over Brazil. Journal of Geophysical Research: Space Physics, 123(3), 2215-2227.

- Kelley, M. C., and Miller, C. A. (1997), Electrodynamics of midlatitude spread F 3. Electrohydrodynamic waves? A new look at the role of electric fields in thermospheric wave dynamics, J. Geophys. Res., 102( A6), 11539– 11547, doi:10.1029/96JA03841.
- Otsuka, Y., Suzuki, K., Nakagawa, S., Nishioka, M., Shiokawa, K., & Tsugawa, A. (2013). GPS observations of medium-scale traveling ionospheric disturbances over Europe. Annales Geophysicae (09927689), 31(2).
- Tsai, L. C., Su, S. Y., & Liu, C. H. (2017). Global morphology of ionospheric F-layer scintillations using FS3/COSMIC GPS radio occultation data. GPS Solutions, 21(3), 1037-1048.
- Syndergaard, S. (2006). COSMIC S4 Data. COSMIC Data Analysis and Archival Center at UCAR. *https://cdaac-www.cosmic.ucar.edu/cdaac/doc/documents/s4_description.pdf*

---

## Author Comment (AC3) · 28 May 2019

Anonymous Reviewer #3

Global Climatology of Equatorial Plasma Bubbles based on GPS Radio Occultation from FormoSat-3/COSMIC by Kepkar et al looks at the seasonal, longitudinal, annual, altitude and local time variations in the occurrence of equatorial plasma bubbles as indicated by the COSMIC S4 index. The authors have failed to highlight the novelty of the work, and as such I cannot recommend it for publication. Their work is very similar to Carter et al 2013, whom they cite in reference to variation with solar activity. However, they have failed to discuss the work in the context of this paper even though Carter et al 2013 looked at the seasonal, longitudinal, annual and local time variations of equatorial plasma bubbles as indicated by the COSMIC S4 index.

➔ We like to thank the reviewer for his time in going through paper and providing constructive inputs that will improve the quality of the paper. We agree that occurrence climatology presented in this paper doesn't explain the similarities and dissimilarities in-detail with the previous studies. In order to make this study worthy of publication, we tried to address all the issues raised.

Indeed, Figure 6 of this paper is very similar to Figure 4 of Carter et al 2013. The years of data used are different. However, the differences and similarities between Figure 6 and the results from Carter et al 2013 are not discussed and it is showing how this work provides something new that is currently missing from the manuscript.

➔ We are pleased to have authors perspective concern to include differences and similarities shown in the Fig. 6 with previous publication including Carter et al. (2013). One of the main differences from the Carter's et al. (2018) paper, that this study includes seasonal occurrence surrounding the solar max year as pointed out by the reviewer and we intend to explain in more detail the difference in occurrence characteristics. In particular, only one maximum above geomagnetic equator is visible across different seasons and regions with the most EPB occurrence in the African sector during March equinox for solar maximum years (2012-2016). It is also known that different measurement techniques incur different occurrence interpretation and therefore we planned to include comprehensive comparison of EPBs based on previous publication (Burke et al. (2004a, 2004b), Gentile et. al (2006), Su et al. (2006), Stolle et al. (2006), Nishioka et. al (2008), Dao et al. (2011), Carter et al. (2013), Xiong et al. (2013), Liu et al. (2016)).

Taking each sentence from the abstract in turn below it can be seen that no new information is currently being highlighted by the authors "The analysis revealed that the F-region irregularities, associated with plasma bubbles occur mainly post sunset close to Earth's geomagnetic equator."- This has been known and written in many papers, including ones such as Sultan 1996 and others that look at the mechanism and growth rate of post sunset plasma bubbles

➔ We would agree with the reviewer's comment, that it has been known that bubbles predominantly occur at the geomagnetic equator. However, we would like to present the maxima of EPB occurrence shifting from the South American sector towards the African sector on a year wise basis, while proceeding the solar maximum year. Subsequently we plan to add this in the manuscript and make changes in the abstract too. In the local time distribution, we already know that plasma bubbles occur post sunset and therefore we

showed brief plot and variation according to the solar activity that provides manifestation for GPS-RO as a complementary method for studying the EPBs as also mentioned by Carter et. al (2013) in his paper.

"Dependence on the solar cycle as well as distinctive seasonal variation is observed when analysed for different years." -Bourke et al 2004 looked at the climatology of plasma bubbles for both low and high solar activity, and Carter et al 2013 looked at the climatology of plasma bubbles using COSMIC S4 as a function of year.

→ We are agreeing with the reviewer comment when it comes to the previous work done by different authors. In this paper, the seasonal variation of EPBs agrees well with the results of Burke et al. (2004) and Su et al. (2006) with peak for equinox in the African sector. However, it differs from the observation made by Carter et al. (2013), where he observes peak for equinox in the American region. This could be because of the study period (2007-2011) which covers solar minimum covered by Carter et al. (2013), while Su. et al. (2006) and this paper used datasets surrounding the solar maximum year. We plan to include this and explain in more-detail the occurrence in region wise seasonal dependence of EPBs.

"In contrast to the other ionospheric remote sensing methods, GPS Radio Occultation technique uniquely personifies the activity of the plasma bubbles based on altitude resolution on a global scale." –As mentioned throughout this review, Carter et al 2013 used COSMIC RO data to look at plasma bubbles, so more is needed to make this a new finding.

→ It is very well known that that plasma bubbles vary with the altitude. Since not many techniques providing altitude resolution are globally spread, therefore those studies are restricted to a particular region. The GPS RO provides the advantage of having altitude resolution when it comes to studies related to the F-region scintillation (Liu et al. (2016)). So this paper intends to give information on a year wise basis, starting from solar minimum year, covering the solar maximum year followed by decreasing solar activity. It also highlights the EPBs detected from GPS-RO extending to greater altitudes and shifting its peak from lower latitudes to the higher latitudes as we proceed towards solar maximum. More information and in-detail explanation is planned to be added in the revised manuscript.

Minor comments: In section 2.1.1 Derivation of amplitude scintillation index it is unclear if the authors have used the provided s4max9sec data and are explaining how it is derived, or if they have used raw data and re-analysed it themselves. If it is the later the reasoning also needs to be made clear to the reader.

→ The amplitude scintillation index is calculated using raw 1 Hz SNR measurements from ionPhs data. The same has been already documented in section 2.1 (Data availability). For better understanding, the used dataset will be also mentioned again in section 2.1.1 (Derivation of amplitude scintillation index) in the revised manuscript. The ionPhs data is preferred over the ScnLv1 datasets (which includes 50 Hz measurements recorded at 1 Hz (Syndergaard (2006))), because the ionPhs profiles have increased number of dataset (i.e by

factor 5) when compared to ScnLv1 datasets along the F-region altitude. Therefore, to reduce the number of interpolated occurrence number, raw SNR measurements are exploited for this study.

It is unclear why the authors have chosen to use the average S4 rather than some occurrence calculation. Averages can be misleading if the distribution between cells varies, or the number of points vary etc. A justification should be added or another way to demonstrate the data should be used.

➔ We are thankful for the reviewer's hint and agree with his suggestion on having some occurrence calculation. Based on his suggestion, we modify the earlier plots showing S4 average with occurrence rate. The occurrence rate is calculated as a ratio of number of profiles having maximum S4 value greater than 0.3 to the total number of profiles in each grid.

In Figures 5 & 6 the captions state that the Figures show the EPB occurrence. In Figure 5 the numbers seem very low for this to be the case and in Figure 6 it is clearly the S4 average again, this is inconsistent, confusing and needs to be fixed.

➔ Thank you for pointing this out. As mentioned in the previous reply, we have replaced the S4 average with the occurrence rate. This will further unify the results based on the occurrence calculation in the analyses. As far as the numbers are concerned in the Figure 5, the occurrence rate is calculated as ratio of number of profiles having maximum S4 value greater than 0.3 for a particular time interval [for e.g 20 hr – 21 hr] to the total number of profiles in each grid for the same time interval [for e.g 20 hr – 21 hr]. The low numbers could be probably because of using low sampling rate data and we plan to include this reason while comparing our results with Carter et al. (2013).

In order to make this work worthy of publication the authors need to carefully discuss the results in the context of similar papers that are currently not included in the discussion of the results. Following this they need to assess and highlight what is new and different to determine if the work is novel.

➔ Finally, we would like to appreciate the reviewer for giving his comprehensive review on this paper. We will surely add and discuss the results with the previous studies in-detail. We will also incorporate the suggestion and hints provided by reviewer in the revised manuscript and improve the quality of the paper for publication.

**References**

- Burke, W. J., C. Y. Huang, L. C. Gentile, and L. Bauer (2004a), Seasonal longitudinal variability of equatorial plasma bubbles, Ann. Geophys., 22,30893098.
- Burke, W. J., L. C. Gentile, C. Y. Huang, C. E. Valladres, and S. Y. Su (2004b), Longitudinal variability of equatorial plasma bubbles observed by DMSP and ROCSAT-1, J. Geophys. Res., 109, A12301, doi:10.1029/2004JA010583.
- Carter, B. A., Zhang, K., Norman, R., Kumar, V. V., and Kumar, S (2013), On the occurrence of equatorial F-region irregularities during solar minimum using radio occultation measurements, Journal of Geophys. Res., 118, 892–904, https://doi.org/10.1002/jgra.50089.

- Dao, E., M. C. Kelley, P. Roddy, J. Retterer, J. O. Ballenthin, O. de La Beaujardiere, and Y.-J. Su (2011), Longitudinal and seasonal dependence of nighttime equatorial plasma density irregularities during solar minimum detected on the C/NOFS satellite, Geophys. Res. Lett., 38, L10104, doi:10.1029/2011GL047046.
- Nishioka, M., A. Saito, and T. Tsugawa (2008), Occurrence characteristics of plasma bubble derived from global ground-based GPS receiver networks, J. Geophys. Res., 13, A05301, doi:10.129/2007JA012605.
- Liu, J. Y., Chen, S. P., Yeh, W. H., Tsai, H. F., & Rajesh, P. K. (2016). Worst-case GPS scintillations on the ground estimated from radio occultation observations of FORMOSAT-3/COSMIC during 2007–2014. Surveys in Geophysics, 37(4), 791-809.
- Gentile, L. C., W. J. Burke, and F. J. Rich (2006), A global climatology for equatorial plasma bubbles in the topside ionosphere, Ann. Geophys., 24,163–172
- Stolle, C., Lühr, H., Rother, M., and Balasis, G (2006), Magnetic signatures of equatorial spread F as observed by the CHAMP satellite, J. Geophys. Res., 111, https://doi.org/10.1029/2005JA011184
- Su SY, Liu CH, Ho HH, Chao CK (2006) Distribution characteristics of topside ionospheric density irregularities: equatorial versus midlatitude region. J Geophys Res 111:A06305. doi:10.1029/2005JA011330
- Syndergaard, S. (2006). COSMIC S4 Data. COSMIC Data Analysis and Archival Center at UCAR. *https://cdaac-www.cosmic.ucar.edu/cdaac/doc/documents/s4_description.pdf*
- Xiong, C., Park, J., Lühr, H., Stolle, C., & Ma, S. Y. (2010). Comparing plasma bubble occurrence rates at CHAMP and GRACE altitudes during high and low solar activity. Annales Geophysicae, 28(9), 1647-1658.

---

## Referee Report (RR1)

As I mentioned at the previous review, this paper does not deliver new science regarding bubbles. I would rate the scientific value of the paper to be medium low. However, I do not object publication of this paper because this paper can be a reference to other researcher who are interested to use the GPS RO data.

Writing was significantly improved compared with the previous version. The paper is readable as is, although it is not a well written paper. The editor may decide whether to request further elaboration of writing. Below I point out some expressions that bothered me.

I'd like to note one thing. The authors considered scintillation as a proxy to detect bubbles. It can be true because scintillations would be caused by bubbles in low latitudes, but there is a limitation. Let's think about the factors that determine the scintillation intensity. The S4 index is determined by the strength of the irregularity. The irregularity strength is a function of the background density. So, the scintillation distribution is biased by the background density. The occurrence rate of bubbles should be the maximum at the magnetic equator because they initiate there. Satellite observations showed the peak occurrence rate at the magnetic equator. But, in Figure 3 in the manuscript, the occurrence rate is not maximum at the magnetic equator. This result is expected because an ionization trough forms at the magnetic equator. The hemispheric, seasonal, and solar cycle dependence of scintillation will also be affected by the variation of the background density with those factors. An explanation about this (the effect of background plasma density on the distribution of scintillation) is necessary.

**Abstract**
I am not a native English speaker, so I cannot thoroughly review English. But "the" may not necessary in front of "bubbles" because Abstract does not describe specific bubbles. The same in other sections. Please check this.

Line 7: The expression "significant distribution" does not make sense. I would write line 6-8 as "The occurrence of bubbles shows a strong dependence on longitude, season, and solar cycle with the peak occurrence rate in the African sector during March equinox during high solar activity."

Page 1 Line 19: "EPBs instigated by plasma irregularities" is not a good expression. Just "EPBs" is good enough.

Page 2 Line 1-3: Simply say that "EPBs appear as plume-like structure in radar observations and emission depletions in airglow images"

Page 2 line 11: Where do the polarization electric fields that cause bubbles develop? Is it E region?

Page 7 line 13-14: "when the polarization electric field shorts E-region conductivity causing a rapid loss of plasma". I do not understand what it meant. Does the polarization electric field determine the E region conductivity? Does the polarization electric field or E region conductivity cause the rapid plasma loss?

Page 11 line 16: The authors may add following references regarding the relationship between PRE and bubbles:

*Anderson, D. N., B. Reinisch, C. Valladares, J. Chau, and O. Veliz (2004), Forecasting the occurrence of ionospheric scintillation activity in the equatorial ionosphere on a day-to-day basis, JASTP, 66, 1567-1572.*

*Fejer, B. G., L. Scherliess, and E. R. de Paula (1999), Effects of the vertical plasma drift velocity on the generation and evolution of equatorial spread F, J. Geophys. Res., 104, 19,859-19,869.*

*Kil, H., L. J. Paxton, and S.-J. Oh (2009), Global bubble distribution seen from ROCSAT-1 and its association with the pre-reversal enhancement, J. Geophys. Res., 114, A06307, doi:10.1029/2008JA013672.*

*Li, G., B. Ning, L. Liu, Z. Ren, J. Lei, and S.-Y. Su (2008), The correlation of longitudenal/seasonal variations of evening equatorial pre-reversal drift and of plasma bubbles, Ann. Geophys., 25, 2571-2578.*

*Su, S. -Y., C. K. Chao, and C. H. Liu (2008), On monthly/seasonal/longitudinal variations of equatorial irregularity occurrences and their relationship with the post-sunset vertical drift velocities, J. Geophys. Res, 113, A05307, doi:10.1029/2007JA012809.*

Page 11 line 18-19: "The seed perturbation along with the altitudinal variation of the EPBs is largely attributed to the PRE." I am confused of the seed perturbation mentioned in this sentence. Do you mean seed perturbations such as AGWs? If it is, how do they have a connection to the PRE?

Page 11 line 25: I am not sure of the meaning of "materialize"

---

## Referee Report (RR2)

Global Climatology of Equatorial Plasma Bubbles based on GPS Radio Occultation from FormoSat-3/COSMIC by Kepkar et al

In general, I am happy with the content changes to this paper compared with the previous submission. The authors have provided an extended discussion in multiple places throughout the paper in order to discuss the results in the context of the literature. It is a dramatic improvement. At this point I still have some comments that should be addressed before the paper is published. Therefore, I recommend minor revisions.

One thing I must address in both the paper and the response to reviewers is the use of gendered pronouns. In the case of a response to an anonymous reviewer the authors should be aware that they don't know the gender of the reviewer and should therefore not use gendered pronouns. In the case of the paper, an example is noted on page 7, line 29 "wherein in his analysis" this is both applying the male gender to all co-authors on the paper and it is a singular term when the paper was authored by more than 1 person. Another example occurs on page 14 line 12 "through his comparative studies" which has the same problems of gender and singular.

I have 2 major comments and multiple minor comments listed below:

Major comments

Firstly, I am concerned about the occurrence in some of the Figures. For example I would expect the solar maximum histogram in Figure 5 to have similar values to the occurrence observed in Figure 4. However, Figure 4 seems to have regions with 20% (or above) occurrence, while Figure 4 only goes to 8% and in general these occurrences seem rather small unless there is a bias towards soundings at the higher latitudes.

While the scientific results in the paper are interesting and the conclusion section focusses the reader on what is new and novel in the work, I still feel that the abstract has missed the key novel findings of the paper. The abstract still appears to focus on confirming what was already known in the literature and not what this paper shows. I think the authors should consider re-writing the abstract to align more with the focus in the conclusions section.

Minor comments

The minor comments can be summed up as the grammar and style of the writing still needs a bit more work.

The authors regularly change the writing style, e.g. in some places it is passive past tense (as science writing should usually be) and then in other places they are using words like we, making it active; some places being active current tense "we know" and sometimes active past tense "In this paper, we" or even active future tense "we can witness". The entire paper should be edited for these style and grammar inconsistencies.

The authors have not defined the acronym GPS

There are some places where the authors have used plural or singular terms incorrectly (in addition to the cases of "his"), for example on page 10, line 9 "maximum occurrence during both equinoxes are observed in Africa and agrees well…"  the author should have "agree" rather than "agrees"

In some places articles are missing e.g. page 11 line 16 "consequence of RTI" should be "consequence of the RTI"

The authors should ensure the correct adjectives are used throughout the paper. In particular, using strong/weak to refer to size should be avoided (particularly since there are places where strong/weak is appropriate to use). E.g. on page 14 line 7 "stronger magnitude" should be "larger magnitude"

There are a few places where the phrasing is odd or wrong e.g. "On hindsight", the phrase is "in hindsight" and I don't understand what the authors mean by "merely detected" on page 14 line 15 (e.g. does it mean detected but nothing else is done with it, detected but it has no effect etc (these are the normal uses of the phrase merely detected) if the authors mean "only just detected" then they should say that, and provide context about what they mean (e.g. only small dips in density observed))

There are also many "hanging" its. In other words, sentences where the "its" is ambiguous. For example, on page 14 line 10 "it justifies" I have no idea what is doing the justifying.

There are many places where changing "than" to "compared with" would make things smoother and add clarity.

The examples listed above are just examples, there are many more instances of these grammar and style problems throughout the paper and the authors should go through the paper and ensure the scientific writing is up to scratch.

---

## Author Response (AR2)

Response to anonymous reviewer #1

As I mentioned at the previous review, this paper does not deliver new science regarding bubbles. I would rate the scientific value of the paper to be medium low. However, I do not object publication of this paper because this paper can be a reference to other researcher who are interested to use the GPS RO data.

Writing was significantly improved compared with the previous version. The paper is readable as is, although it is not a well written paper. The editor may decide whether to request further elaboration of writing. Below I point out some expressions that bothered me.

➔ First of all, we are very grateful to the reviewer for taking out the time and reviewing this paper once again and providing constructive inputs for improving the manuscript. In the following, we plan to address the issues and include the comments and suggestions for further refinement of the paper.

I'd like to note one thing. The authors considered scintillation as a proxy to detect bubbles. It can be true because scintillations would be caused by bubbles in low latitudes, but there is a limitation. Let's think about the factors that determine the scintillation intensity. The S4 index is determined by the strength of the irregularity. The irregularity strength is a function of the background density. So, the scintillation distribution is biased by the background density. The occurrence rate of bubbles should be the maximum at the magnetic equator because they initiate there. Satellite observations showed the peak occurrence rate at the magnetic equator. But, in Figure 3 in the manuscript, the occurrence rate is not maximum at the magnetic equator. This result is expected because an ionization trough forms at the magnetic equator. The hemispheric, seasonal, and solar cycle dependence of scintillation will also be affected by the variation of the background density with those factors. An explanation about this (the effect of background plasma density on the distribution of scintillation) is necessary.

➔ We are thankful to the reviewer for pointing this out, which is interesting and related to this study. The S4 index is indeed determined by the strength of the irregularity which in turn is a function of the background electron density. From the previous studies, it is known that as we proceed towards high solar activity, the equatorial anomaly produces higher values of electron density with maxima at the crest latitudes (J. Aaron, 1982). Furthermore, it evident from various probing techniques that the generation of the plasma bubbles commences at or near the magnetic equator. Based on the equatorial electrodynamics, at the time of sunset, the PRE causes the F-region region to move upward and trigger the Rayleigh-Taylor instabilities which also leads to a reappearance of the equatorial anomaly after the sunset especially during the high solar activity (Basu et al., 2002). As a result, irregularities occur in the region of high ionization density, i.e., crest, in the anomaly region. Such results have been achieved previously using different techniques, which showed strong scintillations in the anomaly latitudes compared to the dip equator.

Abstract

I am not a native English speaker, so I cannot thoroughly review English. But "the" may not necessary in front of "bubbles" because Abstract does not describe specific bubbles. The same in other sections. Please check this.

➔ According to the reviewer's suggestion, we have removed 'the' in front of bubbles and the changes are incorporated in the abstract.

Line 7: The expression "significant distribution" does not make sense. I would write line 6-8 as "The occurrence of bubbles shows a strong dependence on longitude, season, and solar cycle with the peak occurrence rate in the African sector during March equinox during high solar activity."

➔ We have taken the reviewer's suggestion into consideration and the corresponding modifications have been made in the sentence.

Page 1 Line 19: "EPBs instigated by plasma irregularities" is not a good expression. Just "EPBs" is good enough.

➔ Thank you for providing your perspective suggestion for this particular sentence. We have incorporated this changes in the manuscript.

Page 2 Line 1-3: Simply say that "EPBs appear as plume-like structure in radar observations and emission depletions in airglow images"

➔ We are grateful to the reviewer for the ideal reconstruction of the sentence and has been modified in the current manuscript.

Page 2 line 11: Where do the polarization electric fields that cause bubbles develop? Is it E region?

➔ The polarization electric field that develops bubbles occurs in the F-region. However, when seeding of equatorial plasma bubbles by gravity waves are considered, they cause perturbation in the lower thermosphere, i.e., E-region, which then maps it onto the bottom-side of F-region along the magnetic field lines by electrodynamical coupling during the late afternoon (Tsunoda, 2010; Tsunoda, 2015). We agree that in the manuscript, it is not clear, and we intend to include an extended explanation.

Page 7 line 13-14: "when the polarization electric field shorts E-region conductivity causing a rapid loss of plasma". I do not understand what it meant. Does the polarization electric field determine the E region conductivity? Does the polarization electric field or E region conductivity cause the rapid plasma loss?

➔ We understand that this particular sentence conveys a wrong meaning. The polarization electric field neither determines E region conductivity nor cause plasma loss. The modified sentence is: "The rapid depletion of the E-region conductivity and the onset of PRE right after sunset cause the plasma bubble to develop, i.e., ~19:00 LT. This characteristic is evident from the local time occurrence of EPBs shown in Fig. 4 and agrees with the study from Stolle et al., (2006).

Page 11 line 16: The authors may add following references regarding the relationship between PRE and bubbles:

*Anderson, D. N., B. Reinisch, C. Valladares, J. Chau, and O. Veliz (2004), Forecasting the occurrence of ionospheric scintillation activity in the equatorial ionosphere on a day-to-day basis, JASTP, 66, 1567-1572.*

*Fejer, B. G., L. Scherliess, and E. R. de Paula (1999), Effects of the vertical plasma drift velocity on the generation and evolution of equatorial spread F, J. Geophys. Res., 104, 19,859-19,869.*

*Kil, H., L. J. Paxton, and S.-J. Oh (2009), Global bubble distribution seen from ROCSAT-1 and its association with the pre-reversal enhancement, J. Geophys. Res., 114, A06307, doi:10.1029/2008JA013672.*

*Li, G., B. Ning, L. Liu, Z. Ren, J. Lei, and S.-Y. Su (2008), The correlation of longitudenal/seasonal variations of evening equatorial pre-reversal drift and of plasma bubbles, Ann. Geophys., 25, 2571-2578.*

*Su, S. -Y., C. K. Chao, and C. H. Liu (2008), On monthly/seasonal/longitudinal variations of equatorial irregularity occurrences and their relationship with the post-sunset vertical drift velocities, J. Geophys. Res, 113, A05307, doi:10.1029/2007JA012809.*

➔ I would like to thank the reviewer for providing the literature which was very informative and provided a better understanding of the relationship between PRE and bubbles.

Page 11 line 18-19: "The seed perturbation along with the altitudinal variation of the EPBs is largely attributed to the PRE." I am confused of the seed perturbation mentioned in this sentence. Do you mean seed perturbations such as AGWs? If it is, how do they have a connection to the PRE?

➔ We agree with the reviewer that this phrase is confusing. AGWs and PRE are both different and have no connection or relation. The rephrased sentence included in the manuscript is: The altitudinal uplift of EPBs is mainly due to the magnitude of PRE."

Page 11 line 25: I am not sure of the meaning of "materialize"

➔ The meaning of the word 'materialize' is to 'emerge'. Since plasma bubble exists in clusters, therefore this word seems more relatable.

We would once again thank the reviewer for his time in reviewing this manuscript again. We have incorporated all the suggestions and modifications for further refinement of the paper.

In general, I am happy with the content changes to this paper compared with the previous submission. The authors have provided an extended discussion in multiple places throughout the paper in order to discuss the results in the context of the literature. It is a dramatic improvement. At this point I still have some comments that should be addressed before the paper is published. Therefore, I recommend minor revisions.

➔ We would like to thank the reviewer for their time in reviewing this manuscript again and providing insightful inputs. As far as the manuscript is concerned, we are also pleased to learn that the reviewer was content with the modification in the paper. We have addressed the issues raised by the reviewer and incorporated the modifications suggested.

One thing I must address in both the paper and the response to reviewers is the use of gendered pronouns. In the case of a response to an anonymous reviewer the authors should be aware that they don't know the gender of the reviewer and should therefore not use gendered pronouns. In the case of the paper, an example is noted on page 7, line 29 "wherein in his analysis" this is both applying the male gender to all co-authors on the paper and it is a singular term when the paper was authored by more than 1 person. Another example occurs on page 14 line 12 "through his comparative studies" which has the same problems of gender and singular.

➔ Thanking you for pointing this out. We completely agree that this is incorrect, and would further avoid using gendered pronouns while referring to a research paper as listed in the comment.

I have 2 major comments and multiple minor comments listed below:

Major comments

Firstly, I am concerned about the occurrence in some of the Figures. For example I would expect the solar maximum histogram in Figure 5 to have similar values to the occurrence observed in Figure 4. However, Figure 4 seems to have regions with 20% (or above) occurrence, while Figure 4 only goes to 8% and in general these occurrences seem rather small unless there is a bias towards soundings at the higher latitudes.

➔ Fig. 4 shows the local time occurrence of plasma bubble in the grid of 1-hour x 5° latitude. For instance, the maximum occurrence, i.e., ~20 %, is observed in the latitudinal grid of +15° and +20° and around 21:00 local time. However, In Fig. 5, which reveals the comparison with respect to local time each for solar maximum and minimum year, the EPB occurrence is computed taking into account the geographical latitudes between -50° and 50°. Thus, Fig. 5 has a lower occurrence number compared to Fig. 4.

While the scientific results in the paper are interesting and the conclusion section focusses the reader on what is new and novel in the work, I still feel that the abstract has missed the key novel findings of the paper. The abstract still appears to focus on confirming what was already known in

the literature and not what this paper shows. I think the authors should consider re-writing the abstract to align more with the focus in the conclusions section.

➔ As suggested by the reviewer, we agree on the need for re-writing the abstract in line with the scientific results which this paper outlines. Given that, we have re-written the abstract outlining the key findings that will further improve the overall quality of this paper.

Minor comments

The minor comments can be summed up as the grammar and style of the writing still needs a bit more work.

➔ We are thankful to the reviewer for pointing this out, which also is an important factor while drafting a manuscript. Therefore, we have incorporated the changes relating to overall writing style and grammar, as suggested by the reviewer.

The authors regularly change the writing style, e.g. in some places it is passive past tense (as science writing should usually be) and then in other places they are using words like we, making it active; some places being active current tense "we know" and sometimes active past tense "In this paper, we" or even active future tense "we can witness". The entire paper should be edited for these style and grammar inconsistencies.

➔ We have included the suggestion and also considered re-writing the manuscript in the passive voice at the same time ensuring the consistency.

The authors have not defined the acronym GPS

➔ We would like to thank you for bringing this to our notice and included the acronym for GPS which is the Global Positioning System in the revised manuscript.

There are some places where the authors have used plural or singular terms incorrectly (in addition to the cases of "his"), for example on page 10, line 9 "maximum occurrence during both equinoxes are observed in Africa and agrees well…" the author should have "agree" rather than "agrees"

➔ We are aware of this error and have tried rectifying the plural and singular terms used in the manuscript.

In some places articles are missing e.g. page 11 line 16 "consequence of RTI" should be "consequence of the RTI"

➔ We have incorporated the changes related to missing articles, as suggested by the reviewer in the revised manuscript.

The authors should ensure the correct adjectives are used throughout the paper. In particular, using strong/weak to refer to size should be avoided (particularly since there are places where strong/weak is appropriate to use). E.g. on page 14 line 7 "stronger magnitude" should be "larger magnitude"

➔ We have noted this, and the same have been incorporated to rectify the correct usage of the adjectives.

There are a few places where the phrasing is odd or wrong e.g. "On hindsight", the phrase is "in hindsight" and I don't understand what the authors mean by "merely detected" on page 14 line 15 (e.g. does it mean detected but nothing else is done with it, detected but it has no effect etc (these are the normal uses of the phrase merely detected) if the authors mean "only just detected" then they should say that, and provide context about what they mean (e.g. only small dips in density observed))

➜ We have addressed the raised issue in the manuscript by reconstructing a proper phase.

There are also many "hanging" its. In other words, sentences where the "its" is ambiguous. For example, on page 14 line 10 "it justifies" I have no idea what is doing the justifying.

➜ We have reconstructed a clearer sentence in the manuscript taking into consideration the reviewer's suggestion.

There are many places where changing "than" to "compared with" would make things smoother and add clarity.

➜ According to the reviewer's suggestion, we have made the changes in the revised manuscript.

The examples listed above are just examples, there are many more instances of these grammar and style problems throughout the paper and the authors should go through the paper and ensure the scientific writing is up to scratch.

➜ We would like to thank the reviewer for the in-depth review of this manuscript. We included all the suggestions and modifications asked by the reviewer and have carefully re-written the paper taking into consideration the writing style and grammar, which hopefully stands up to the standards of scientific writing.

Changes suggested by:

1) Reviewer #1 – Magenta
2) Reviewer #2 - Blue

[revised manuscript text omitted]

---

## Author Response (AR3)

**Anonymous reviewer #1**

This paper may be publishable because it provides a reference to the researchers who are interested in the use of the GPS-RO data for the study of ionospheric irregularities.

This paper still has lots of problems in writing. I do not expect perfect English, but this current version does not meet the minimum standard of the international journal. Below I mention just some of them at the beginning of the paper.

➔ **We would like to thank the reviewer for his time in going through the paper and providing constructive inputs to improve the quality of the current manuscript. We have considered reviewer's valuable suggestions and comments and included them in our revised manuscript.**

Line 1-2: Grammatically, "which" after comma takes the whole sentence, but here "which" meant bubbles. The authors did not use the GPS-RO technique to study bubbles or their global distribution; the authors used the scintillation data provided by the GPS-RO technique for the study of bubbles.

➔ **We have noted this and made subsequent changes in the manuscript and also agree on the reviewer's comment on the scintillation data from the GPS-RO technique for the study of bubbles and therefore included reviewer's suggestion in the abstract.**

Line 5: I do not know the meaning of "biased". Here again, "which" was not used correctly.

➔ **We have replaced the word "biased" with "influenced" in the manuscript and also tried to correct the "which" that was used incorrectly.**

Line 6 "Moreover" and Line 7 "Furthermore" are not necessary.

➔ **According to review's suggestion, we have removed the words in the revised manuscript.**

Line 10: "which appeared in congruence with" is not a good expression. I would say "which is consistent with"

➔ **We are grateful to the reviewer for point this out and providing with a suggestion that we have included in the revised manuscript.**

Line 16 "large". It is a vague expression. What size is "large"?

➔ **We have reconstructed a clearer sentence in the manuscript taking into consideration reviewer's suggestion.**

Line 18: I do not know the meaning of "serious implications".

➔ **In the revised manuscript, we have removed "serious implication" and replaced it with "attritions".**

I did not notice the small magnitude of the occurrence probability (Figure 3), but the other reviewer correctly pointed out. Strong scintillations are nighttime phenomena. Then, I would calculate the occurrence probability using the data at night instead of using the data at all local times. That's what people usually do.

➔ **According to reviewer's suggestion, we have focussed on night time occurrence and calculated the occurrence probability by restricting the local time between 18 and 06 LT.**

Strong scintillations are nighttime phenomena, then I would make plots of the LT distribution making midnight be at the middle of plots. In other words, I would make x-range in Figure 4 and other plots from 12 to 12 LT (or 18 – 06 LT) instead of from 0 to 24 LT. Then, we can see the continuous variation of the bubble distribution with LT. This is what people usually do.

➔ **As per reviewer's suggestion, we have changed the x-range in the Figure 4 from 12 to 12 LT to have continuous variation of the bubble distribution with midnight at the middle of plot.**

**We would like again to thank the reviewer for his in-depth review. Taking into consideration all the suggestions and inputs recommended by the reviewer, we are hopeful that this manuscript stands up to the standards of scientific writing.**

**Anonymous reviewer #2**

**We would like to thank the reviewer for his time in reviewing this manuscript and immensely grateful for the encouragement on the manuscript.**

[revised manuscript text omitted]

---

## Author Response (AR4)

We would like to thank the topical editor and all the referees for their valuable inputs and suggestion for improving the paper. We have considered all the recommendation and tried best to address the changes and modification that were requested.

[revised manuscript text omitted]